# Examining the effectiveness of parental strategies to overcome bedwetting: an observational cohort study

Mariusz T Grzeda,[1] Jon Heron,[1] Kate Tilling,[1] Anne Wright,[2] Carol Joinson[1]

## ABSTRACT

**Objective** To examine whether a range of common strategies used by parents to overcome bedwetting in 7½-year-old children (including lifting, restricting drinks before bedtime, regular daytime toilet trips, rewards, showing displeasure and using protection pants) are effective in reducing the risk of bedwetting at 9½ years.

**Design** Prospective cohort study.

**Setting** General community.

**Participants** The starting sample included 1258 children (66.7% boys and 33.2% girls) who were still bedwetting at 7½ years.

**Outcome measure** Risk of bedwetting at 9½ years.

**Results** Using propensity score-based methods, we found that two of the parental strategies used at 7½ years were associated with an increased risk of bedwetting at 9½ years, after adjusting the model for child and family variables and other parental strategies: lifting (risk difference=0.106 (95% CI 0.009 to 0.202), ie, there is a 10.6% (0.9% to 20.2%) increase in risk of bedwetting at 9½ years among children whose parents used lifting compared with children whose parents did not use this strategy) and restricting drinks before bedtime (0.123 (0.021 to 0.226)). The effect of using the other parental strategies was in either direction (an increase or decrease in the risk of bedwetting at 9½ years), for example, showing displeasure (−0.052 (−0.214 to 0.110)). When we re-analysed the data using multivariable regression analysis, the results were mostly consistent with the propensity score-based methods.

**Conclusion** These findings provide evidence that common strategies used to overcome bedwetting in 7½-year-olds are not effective in reducing the risk of bedwetting at 9½ years. Parents should be encouraged to seek professional advice for their child's bedwetting rather than persisting with strategies that may be ineffective.

[1]School of Social and Community Medicine, University of Bristol, Bristol, UK
[2]Evelina London Children's Hospital, St Thomas' Hospital, London, UK

**Correspondence to**
Dr Carol Joinson; carol.joinson@bristol.ac.uk

## Strengths and limitations of this study

► Major strengths of this study are the availability of prospective data on bedwetting and a wide range of covariates associated with both bedwetting and the parental strategies in a large birth cohort.

► Use of propensity score-based methods with observational data makes it easier to assess whether observed confounding has been adequately eliminated.

► We did not separately examine whether the parental strategies are differentially effective for children with non-monosymptomatic and monosymptomatic enuresis.

► We did not have information on the onset or duration of the strategies to overcome bedwetting and, therefore, we were only able to assess strategies that were reported by parents as being used currently.

► There were very small numbers of parents using medication or bedwetting alarms, so we were unable to examine the effectiveness of these interventions in this study.

## INTRODUCTION

Attainment of bladder control is a major milestone in child development that marks the end of a period of toilet training that is sometimes prolonged and stressful for parents and children. Enuresis is the term used by the International Children's Continence Society to describe bedwetting in children aged 5 years or older after ruling out organic causes.[1] It is most common for children to become dry during the day before they remain dry at night,[2] with children usually attaining night-time bladder control between the ages of 4 and 6 years.[3] Although a significant proportion of children continue to wet the bed at school age, only a small proportion wet the bed at least twice a week, the frequency required for Diagnostic and Statistical Manual of Mental Disorders, fifth edition (DSM-V) diagnosis of enuresis. For example, bedwetting was reported in 15.5% of 7½-year-olds in the Avon Longitudinal Study of Parents and Children (ALSPAC), but only 2.6% wet the bed twice a week or more at this age.[4] With increasing age, bedwetting becomes more socially unacceptable and is often met with intolerance and frustration from parents.[5] Bedwetting also places considerable practical and financial burdens on the family in terms of the extra workload of washing bed linen and the cost of protective pants.[6] The psychosocial implications of bedwetting among school-age children include worries about

participating in sleepovers and school trips, fear of detection, teasing from peers, a sense of being different from others, emotional distress and low self-esteem.[6] Achieving bladder control is, therefore, important for a child's health and well-being.

It is a common belief among parents that bedwetting will eventually resolve with age and, as a result, many parents delay seeking treatment for bedwetting until it is having a considerable impact on the child and family.[7] There is evidence from randomised and quasi-randomised trials that treatment of bedwetting with an alarm or medication can be effective,[8 9] but many parents are unaware that effective treatments for bedwetting are available.[10 11] Before seeking medical advice, parents often employ a range of simple strategies aimed at overcoming bedwetting, the most common being restricting drinks before bed, lifting (removing the sleeping child from bed to empty the bladder in the toilet or potty), rewarding for being dry, regular daytime toilet trips, using protection pants and showing displeasure.[10] Caldwell *et al*[12] conducted a systematic review of randomised (or quasi-randomised) trials of simple behavioural interventions that are often used by parents as first-line interventions for bedwetting. The studies compared the interventions with an appropriate comparison group, comprising no active treatment, other behavioural interventions, or drugs either alone or in combination with other interventions. They found evidence that simple behavioural interventions (eg, rewards, lifting) were more effective in promoting dryness than no intervention, but were inferior to treatments such as alarms and medication. The authors of the review noted, however, that most trials were small and of poor methodological quality, and caution should be exercised in interpreting the findings.

Although the best method for drawing causal inferences about interventions is a randomised controlled trial (RCT), this type of study design is not always feasible or ethical. When an RCT is not possible, observational studies are the only available type of evidence. Causality is difficult to assess in such studies because of the possibility of confounding—where one or more variable(s) influence both the exposure and the outcome, and thus it appears that there is a causal link between them. One method to assess the effectiveness of an intervention while adjusting for known and measured confounders is by using methods based on propensity scores. The aim of this study is to apply propensity score-based methods to observational data from a birth cohort to examine the effectiveness of a range of parental strategies aimed at overcoming childhood bedwetting and to compare this with results using regression methods.

## METHODS
### Participants
The sample comprised participants from the ALSPAC. Detailed information about ALSPAC is available on the study website (http://www.bristol.ac.uk/alspac), which includes a fully searchable dictionary of available data (http://www.bris.ac.uk/alspac/researchers/data-access/data-dictionary). Pregnant women resident in the former Avon Health Authority in south-west England, having an estimated date of delivery between 1 April 1991 and 31 December 1992 were invited to take part, resulting in a cohort of 14 541 pregnancies and 13 973 singletons/twins (7217 boys and 6756 girls) alive at 12 months.[13] Ethical approval for the study was obtained from the ALSPAC Law and Ethics committee and local research ethics committees. Written informed consent was obtained after the procedure(s) had been fully explained.

Bedwetting at 7½ and 9½ years: The starting sample for this study comprised children who were still bedwetting at 7½ years. Parents were asked "How often does your child wet him/herself during the night?" and for both questions were given the response options 'Never', 'Occasional accident but less than once a week', 'About once a week', '2–5 times a week', 'Nearly every day' and 'More than once a day'. A total of 8151 parents responded to this questionnaire and 1258 children (15.4%) were still wetting the bed at 7½ years, comprising 840 (66.7%) boys and 418 (33.2%) girls. Of the 1258 children, 215 (69.8% boys and 30.2% girls) wet the bed 'at least twice a week'. At age 9½ years, the questions on frequency of bedwetting were repeated. Bedwetting data were provided for 8101 children and 788 (9.7%) wet the bed at this age (120 of these children wet the bed at least twice a week). The proportion of children with bedwetting at 7½ years whose parents provided data at age 9½ was 83% (1045 out of 1258) and 47% (n=493) were still wetting the bed at this age, comprising 357 (72.4%) boys and 136 (27.6%) girls.

### Parental strategies to overcome bedwetting at 7½ years
A questionnaire was administered to parents when the study children were aged 7½ that contained a list of strategies aimed at overcoming bedwetting that were originally elicited from parents.[10] Parents were asked: "Which of the following methods have you tried in the past or are you using now to try and help your child stop wetting the bed?" We restricted our analysis to strategies that parents reported 'using now' because there was no information on the onset or duration of strategies used in the past. The questionnaire explained that these strategies are not necessarily effective in overcoming bedwetting.

### Rationale for using propensity score methods in this study
Baseline characteristics of children whose parents used particular strategies to overcome bedwetting (referred to as 'treated' hereafter) might be expected to differ systematically from those whose parents did not use the strategy ('non-treated'), giving rise to confounding. It is, therefore, necessary to account for these differences when estimating the effect of the parental strategy ('treatment') on bedwetting. This could be achieved through multivariable regression (adjusting for differences in baseline variables between treated and untreated participants). Propensity score matching (PSM) has been proposed as

a more appropriate method to minimise confounding when estimating treatment effects using observational data.[14] A detailed description of our rationale for using propensity score-based methods in our study is provided in the online supplementary materials.

## Statistical analysis

We used PSM-based methods to assess the effectiveness of the parental strategies to overcome bedwetting and compared this with estimates derived using logistic regression. Both of these analyses aimed to estimate the effect of each parental strategy (used by parents of children who wet the bed at 7½ years) on risk of bedwetting at 9½ years.

### Propensity score-based model

We assessed the effectiveness of each strategy by examining the difference in risk of bedwetting at 9½ years between children receiving the parental strategy ('treated') versus those children not receiving the strategy ('non-treated'). The estimate we computed was the *average treatment effect for the treated* (ATT)[15] (see online supplementary material for an explanation of our choice of ATT as the measure of treatment effect). In order to put the results of PSM analyses in context, we report estimates obtained both from unadjusted models (without any confounders) and from models adjusted for the set of measured confounders.

We implemented the inverse probability weighting (IPW) method using the propensity score because there is evidence that this method is relatively less biased than other methods based on propensity scores.[15 16] Aside from this, in the reweighting based on propensity scores, as opposed to matching on propensity scores, no case is excluded from the analysis. The baseline variables we selected into the propensity score model were those that were associated with the treatments (parental strategies used at 7½ years) and outcome (bedwetting at 9½ years).

We used a two-stage procedure to derive the list of baseline variables to include in the model. First, we examined two logistic regression models: in model 1, the outcome variable was predicted from all theoretically relevant variables relating to children and their environment (see online supplementary table S1 in the supplementary materials for the list of all variables included). In model 2, the parental strategy was the outcome variable. In the second stage, we applied a threshold for selection of variables to the propensity score model. The variables we included were those that had associations (expressed in terms of odds ratios) with the outcome variables in both models of <0.90 or >1.20, thus allowing us to detect at least 'weak' associations.[17] We estimated models with and without adjustment for the confounders. A detailed list of the confounding variables is provided in online supplementary table S1. Parents frequently used multiple strategies concurrently (indicated by strong correlations between subsets of parental strategies—Table 2). For this reason, at the second stage of analysis, we repeated the IPW models for every strategy adjusting for confounders

and strategies that were correlated (correlation >0.45; see Table 2) with the particular strategy used as the outcome variable in the model. Before analysing the treatment effects, we assessed the adequacy of the propensity score model using diagnostic procedures detailed in the online supplementary material (see online supplementary tables S1–S7, figures S1–S6).

### Logistic regression model

We also examined the effectiveness of the parental strategies to overcome bedwetting by using multivariable logistic regression to adjust for confounders. We applied logistic regression models to the same set of baseline variables used in the propensity score model and reported the results in the form of adjusted risk differences (rather than ORs) for ease of comparison with the results from the propensity score-based model. We estimated risk differences through the STATA *logit* command followed by *adjrr* command.[18] The latter command returned estimates equivalent to average treatment effects (see supplementary materials). In order to obtain robust estimates of the coefficients and their standard errors, we used loops developed in STATA in exactly the same way as for propensity score model.

## Missing data

Many of the potential confounders we considered had some missing data, thus reducing the sample size available for analyses. The number of missing cases on confounding variables ranged from 0 (gender) to 659 (mother's social class based on occupation) (see online supplementary table S1). There were no missing data on exposure (treatment) variables. There were 213 missing cases on the outcome variable (bedwetting at 9½ years).

The exact numbers describing the amount of missing data per variable is presented in the supplementary materials (see online supplementary table S1). To deal with the missing data, we used multiple imputation by chained equations[19] within ICE (Imputation by Chained Equations) STATA package V.14.[20] Full details of the methods used are provided in the supplementary materials.

## RESULTS

Table 1 describes the characteristics of the study participants.

Figure 1 shows the prevalence of each strategy used by parents of 7½-year-old children and the total number of strategies employed by parents. The most common strategies for overcoming bedwetting were restricting drinks, rewarding and lifting. Only a very small number of parents used medication or bedwetting alarms, so we were unable to examine the effectiveness of these interventions. Over 50% of parents did not use any of the strategies to overcome bedwetting, while over 26% of parents used only one strategy. Table 2 shows the associations between the strategies. The strongest correlations were found between lifting and restricting drinks and between regular daytime toilet trips and rewards.

**Table 1** Characteristics of the study participants at 7½ and 9½ years

| | 7½ years (n=1258) | 9½ years (n=1045) |
|---|---|---|
| Gender | | |
| Female | 33.2% | 32.8% |
| Male | 66.8% | 67.2% |
| n | 1258 | 1045 |
| Social class* | | |
| Non-manual (professional, managerial, skilled professions) | 82.0% | 82.3% |
| Manual (partly or unskilled occupations) | 18.0% | 17.7% |
| n | 1094 | 923 |
| Home ownership* | | |
| Owner/occupier | 85.5% | 87.5% |
| Rented accommodation | 14.5% | 12.5% |
| n | 1112 | 952 |
| Car access* | | |
| Yes | 91.1% | 91.8% |
| No | 8.9% | 8.2% |
| n | 1132 | 965 |
| Maternal education* | | |
| A-level or above | 43.9% | 46.9% |
| O-level | 33.6% | 33.4% |
| Certificate of secondary school/vocational/none | 22.5% | 19.7% |
| n | 1226 | 1021 |

*These variables were derived from responses to a questionnaire completed by mothers during the antenatal period.

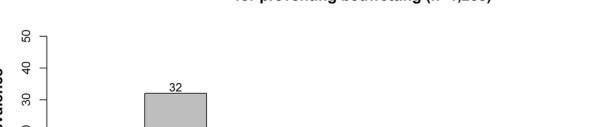

**Fig.1a Prevalence (%) of parental strategies for preventing bedwetting (n=1,258)**

**Fig.1b Prevalence (%) of children with parents employing a given number of strategies (n=1,258)**

**Figure 1** Prevalence of parental strategies for overcoming bedwetting (A) and prevalence of children with parents employing a given number of strategies (B).

Table 3 shows the average treatment effects on treated for each parental strategy used at 7½ years on risk of bedwetting at 9½ years derived from the propensity score-based analysis. The table also shows the average treatment effects derived from the logistic regression model. The coefficients in table 3 are estimated differences between the risk of bedwetting at 9½ years after receiving given 'treatment' (parental strategy) and the risk of bedwetting if they had remained 'untreated'. Figure 2 displays the results of both analyses as forest plots.

## Propensity score-based results (inverse probability weighting)

The unadjusted propensity score-based analysis shows that the parental strategies used at 7½ years are associated with an increase in the risk of bedwetting at 9½ years. Showing displeasure was the only strategy that was also associated with a small decrease in the risk of bedwetting in the unadjusted model. After applying the propensity score model adjusting for the child and family variables and then further adjusting for the other strategies, the adjusted treatment effects provided evidence that lifting and restricting drinks are associated with an increase in the risk of bedwetting at 9½ years. The adjusted results for the other strategies indicated that the effect could be in either direction (either an increase or decrease in the risk of bedwetting).

## Logistic regression results

The results obtained from the logistic regression analysis were mostly consistent with the analysis using the propensity score-based methods, that is, both lifting and restricting drinks were associated with an increase in the risk of bedwetting at 9½ years in the fully adjusted model. Using rewards was also associated with an increased risk of bedwetting at 9½ years.

## DISCUSSION

We examined a range of common strategies used by parents to overcome bedwetting and found that when these strategies were used with 7½-year-old children who wet the bed, they were not effective in reducing the risk of bedwetting at 9½ years. Parental strategies including lifting and restricting drinks before bedtime were associated with an increased risk of subsequent bedwetting. These were among the most common parental strategies used to overcome bedwetting in our study, and there is evidence in a review that these strategies are frequently used by parents across different countries.[10]

This is the first study, to our knowledge, to apply propensity score-based methods to examine the effectiveness of parental strategies to overcome bedwetting using observational data from a large birth cohort. Austin[14] discusses several reasons for preferring the use of propensity score-based methods to regression models when estimating treatment effects using observational data. In particular, it is easier to assess whether observed confounding has been adequately eliminated using propensity score-based

**Table 2** Associations between the parental strategies used to overcome bedwetting in children at 7½ years (n=1258)

| | Lifting child out of bed to use toilet | Restricting drinks before bedtime | Regular daytime toilet trips | Rewarding child for being dry | Showing displeasure when child wets bed | Using night-time protection pants/ nappies | Medication for bedwetting | Bedwetting alarm |
|---|---|---|---|---|---|---|---|---|
| Lifting | 1.000 | | | | | | | |
| Restricting drinks | 0.509 | 1.000 | | | | | | |
| Daytime toilet trips | 0.295 | 0.413 | 1.000 | | | | | |
| Rewards | 0.444 | 0.407 | 0.523 | 1.000 | | | | |
| Showing displeasure | 0.236 | 0.456 | 0.323 | 0.345 | 1.000 | | | |
| Protection pants | −0.038 | −0.151 | 0.231 | 0.249 | 0.040 | 1.000 | | |
| Medication | 0.238 | 0.164 | 0.416 | 0.423 | 0.016 | 0.296 | 1.000 | |
| Bedwetting alarm | 0.050 | −0.099 | 0.230 | 0.492 | −0.070 | 0.230 | 0.113 | 1.000 |

The associations between strategies are tetrachoric correlation coefficients. These are correlation coefficients of binary variables. The tetrachoric correlation coefficient provides an estimate of what the correlation would be if the variables were measured on a continuous scale. The size of the tetrachoric correlation coefficient can be interpreted in the same way as a correlation coefficient between two continuous variables, that is, '0' indicates no correlation and '1' indicates perfect correlation.
All coefficients were computed on the baseline sample.

methods. It is important to note some caveats associated with PSM-based methods, and these are discussed in the supplementary materials. Another major strength of this study is the availability of a wide range of confounders in the ALSPAC dataset that are associated with bedwetting and with the parental strategies to overcome bedwetting. We did not separately examine whether the parental strategies are differentially effective for children with non-monosymptomatic enuresis (bedwetting with daytime wetting and/or lower urinary tract symptoms)[21] and monosymptomatic enuresis (bedwetting without these symptoms).[22] There is evidence that frequent childhood bedwetting that is accompanied by daytime wetting is less likely to resolve with age.[23 24] If we had compared children with monosymptomatic and non-monosymptomatic enuresis, this would necessarily have reduced the number of cases included in the analysis, resulting in serious problems with estimation of the propensity score-based models. We did, however, include a range of covariates in the analyses including frequency of bedwetting (high frequency=twice or more/week), daytime wetting, urgency and voiding postponement. All of these variables were important confounders in most of the models. We did not have information on the onset or duration of the strategies to overcome bedwetting and, therefore, we were only able to assess strategies that were reported by parents as being used currently. Due to this being a community-based, rather than a clinical, sample, there were very small numbers of parents using medication or bedwetting alarms. We were, therefore, unable to examine the effectiveness of these interventions in this study. Additionally, there was no information on which strategies parents initiated compared with those recommended by healthcare workers or other sources (eg, the child's grandparents).

Caldwell *et al*[12] concluded from their systematic review that simple interventions such as lifting (or waking) could initially be tried as strategies to overcome bedwetting since such methods are 'safe and are better than doing nothing'. However, they cautioned that most of the trials they reviewed were small and of poor methodological quality.[12] Our findings are consistent with the current advice given in the National Institute for Health and Care Excellence(NICE) guidelines stating: 'Neither waking nor lifting children and young people with bedwetting, at regular times or randomly, will promote long-term dryness'.[25] Although lifting appears to be sensible for promoting night-time dryness, its effectiveness as an intervention for reducing or stopping bedwetting has previously been questioned. It has been suggested that lifting may inadvertently maintain bedwetting since this strategy encourages the child to empty the bladder without fully waking and, therefore, children are not learning to waken to the sensation of a full bladder.[10 26] However, we are not aware of any studies that have formally tested this proposed mechanism. The effectiveness of lifting requires further investigation in studies that are able to distinguish between children with more severe and less severe bedwetting. Lifting the child from their bed to pass urine can occur with or without waking the child, but this distinction was not possible in our study since parents were not specifically asked whether they

**Table 3** Estimated differences in the risk of bedwetting at 9½ years in children whose parents used each strategy compared with those who did not use the strategy

| | Average treatment effect on treated (95% CI) based on inverse probability weighting using propensity score | | | Average treatment effect/adjusted risk difference (95% CI) based on logistic regression analysis | | |
|---|---|---|---|---|---|---|
| | Empty (unadjusted) model* | Model adjusted for child and family variables† | Model adjusted for child and family variables and other parental strategies‡ | Empty (unadjusted) model§ | Model adjusted for child and family variables¶ | Model adjusted for child and family variables and other parental strategies** |
| Lifting | 0.251 (0.167 to 0.335)†† | 0.148 (0.059 to 0.237)†† | 0.106 (0.009 to 0.202)†† | 0.251 (0.167 to 0.335)†† | 0.172 (0.081 to 0.263)†† | 0.125 (0.029 to 0.221)†† |
| Restricting drinks | 0.173 (0.110 to 0.236)†† | 0.138 (0.071 to 0.205)†† | 0.123 (0.021 to 0.226)†† | 0.173 (0.110 to 0.236)†† | 0.147 (0.083 to 0.211)†† | 0.112 (0.045 to 0.180)†† |
| Daytime toilet trips | 0.198 (0.100 to 0.296)†† | 0.088 (−0.024 to 0.201) | 0.068 (−0.047 to 0.182) | 0.198 (0.100 to 0.296)†† | 0.087 (−0.024 to 0.198) | 0.057 (−0.056 to 0.171) |
| Rewards | 0.281 (0.201 to 0.361)†† | 0.141 (0.049 to 0.233)†† | 0.088 (−0.014 to 0.191) | 0.281 (0.201 to 0.361)†† | 0.158 (0.062 to 0.254)†† | 0.143 (0.044 to 0.242)†† |
| Showing displeasure | 0.054 (−0.091 to 0.199) | 0.009 (−0.146 to 0.164) | −0.052 (−0.214 to 0.110) | 0.054 (−0.091 to 0.199) | 0.009 (−0.134 to 0.151) | −0.047 (−0.187 to 0.093) |
| Protection pants | 0.290 (0.182 to 0.398)†† | −0.010 (−0.161 to 0.141) | −0.010 (−0.161 to 0.141) | 0.290 (0.182 to 0.398)†† | 0.025 (−0.134 to 0.184) | 0.025 (−0.134 to 0.184) |

The estimates provided in this table are average treatment effects for each strategy. They are risk differences, that is, estimated differences between the risk of bedwetting at 9½ years after receiving given 'treatment' (ie, parental strategy) and the risk of bedwetting if they had remained 'untreated'. We provide examples of how to interpret the risk differences below:

(1) The risk difference for 'restricting drinks' (0.123 (95% CI 0.021 to 0.226)) shows that there is a 12.3% (2.1% to 22.6%) increase in risk of bedwetting at 9½ years among children whose parents used restricting drinks compared with children whose parents did not use this strategy.

(2) The risk difference for 'showing displeasure' (−0.052 (−0.214 to 0.11)) shows that there is a 5.2% reduction in the risk of bedwetting at 9 years among children whose parents show displeasure, but the 95% CI indicates that this result could be in either direction (between a 21% reduced risk and 11% increased risk).

*Empty model* column for propensity score-based methods analysis shows unadjusted differences in risk of bedwetting between 'treated' and 'untreated' children. These coefficients are equivalent to bivariate regressions including bedwetting at 9½ years as an outcome variable and each strategy as a single predictor.

†*Model adjusted for child and family variables* represents average treatment effects on treated children. These coefficients are estimated differences in risks of bedwetting in weighted samples. These are differences in risks adjusted for child and family variables that accounted for the differences between treated and untreated groups and between those with and without bedwetting at age 9½. The list of child and family variables was derived separately for every strategy on the basis of ORs from regression analyses providing evidence of an association (ORs <0.90 or >1.20) both with the given strategy and with bedwetting at age 9½. A detailed list of child and family variables included in the model for each parental strategy is provided in the online supplementary material (see online supplementary table S2–S7).

‡*Model adjusted for child and family variables and other parental strategies.* This is the column with coefficients adjusted for the child and family variables and for other parental strategies that were highly correlated (coefficient of tetrachoric correlation >0.45) with given strategy.

§*Empty model*: results of univariable logistic regression analyses including bedwetting at 9½ years as an outcome variable and the given strategy as a single predictor. To ensure direct comparisons with the output of the analysis using propensity score-based methods, the results of regression analyses are expressed here in terms of risk differences instead of ORs.

¶*Model adjusted for child and family variables.* This includes estimated differences in risk of bedwetting for given strategy obtained from logistic regression, also including child and family variables.

**Model adjusted for child and family variables and other parental strategies.* Risk differences for given strategy obtained from logistic regression including child and family variables and other strategies associated with the strategy under examination.

††Indicates risk differences that provide evidence for an *increase* in bedwetting associated with the parental strategy.

woke the child when lifting them to use the toilet or potty. Restricting drinks 1 hour before bedtime is often recommended especially drinks with diuretic properties[27] and drinks containing caffeine.[25] Current NICE guidelines state that adequate daily fluid intake and using the toilet at regular intervals throughout the day are important in the management of bedwetting.[25] We did not find evidence that encouraging regular daytime toilet trips resulted in a decreased risk of bedwetting at 9½ years, but

our findings are based on parental reports rather than a detailed diary of toileting and fluid intake completed during treatment. Using positive (rewards) or negative reinforcements (showing displeasure) as strategies aimed at overcoming bedwetting may be problematic because rewarding outcomes that the child has no control over (ie, a dry night) could lead children to feel that they have failed if they continue to wet the bed.[28] It is recommended to reward steps towards night-time dryness that the child

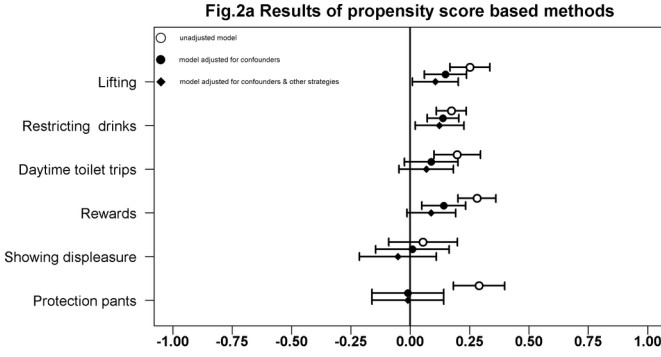

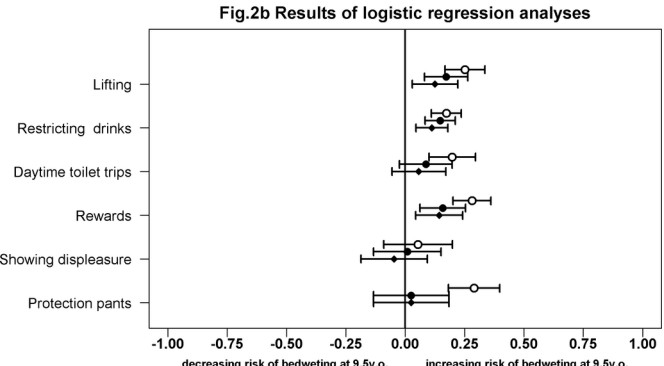

**Figure 2** Estimated differences in the risk of bedwetting at 9½ years in children whose parents used each strategy compared with those who did not use the strategy: results of the propensity score-based model (A) and the logistic regression analysis (B).

*does* have control over (eg, going to the toilet before bed without prompting, drinking and passing urine at regular intervals throughout the day, helping to change sheets).[25] Protection pants (nappies, pull-ups) are often recommended as a useful short-term solution to supporting children during school trips and sleepovers. However, health professionals do not recommend prolonged use of protection pants because this may not help children to become dry in the long term. Instead, parents are advised to consider using alternative bed protection such as waterproof sheets.[25] A trial of at least two nights in a row without protection pants is recommended for children under 5 years who are still bedwetting, but have been dry during the day for more than 6 months.[25]

The majority of children experience a natural resolution of bedwetting with increasing age (spontaneous resolution rate is estimated to be 15% per year).[29] Parents and clinicians, therefore, often adopt a '*wait and see*' approach to childhood bedwetting due to the common belief that it will resolve with age.[7] There is increasing evidence, however, that children who experience frequent bedwetting and those who have bedwetting with additional daytime bladder symptoms are at risk of their problems persisting into adolescence. A cross-sectional study of children aged 5–19 years found a greater proportion of frequent bedwetting (≥3 wet nights per week) in older (11–19 years) compared with younger children (5–10 years).[23] A prospective study found that

children with persistent bedwetting with accompanying daytime wetting had a higher chance of becoming adolescents with bedwetting compared with those who had bedwetting alone in childhood.[24] These findings suggest that parents of children with these patterns of bedwetting should seek early advice from a health professional. In the ALSPAC cohort, however, only 31.9% of parents of 7½-year-old children with bedwetting sought professional help.[10] It was previously common practice not to offer treatment for bedwetting until the child was aged at least 7 years. However, the current NICE guidelines for enuresis recommend that children over the age of 2 years, with ongoing wetting problems, both day and night, who are showing appropriate toileting awareness and behaviour, should be considered for assessment and investigation to exclude a specific medical problem.[25] If initial advice and support does not lead to resolution of bedwetting, NICE recommend that first-line treatment options should be offered to all children with bedwetting, particularly those whose families find the management of bedwetting burdensome and request help. Infrequent bedwetting at age 5 is not uncommon, but if children are still frequently wetting the bed at this age, they may need professional help to become dry at night. There is evidence that treating childhood bedwetting is cost-effective compared with leaving it untreated.[25] Successful treatment can result in children having improved self-esteem[30 31] and quality of life.[32]

This study adds further weight to the importance of encouraging parents to seek professional help for their child's bedwetting, rather than persisting with strategies that may be ineffective. Replication of our findings in other samples would provide further reassurance of our findings. Future research is needed to determine whether there are risk factors in early childhood that predict the continuation of bedwetting at school age. Early identification of the underlying causes of a child's bedwetting, including overactive bladder, nocturnal polyuria and/or problems with sleep and arousal, could help to ensure that appropriate treatment is given.

**Acknowledgements** This study is based on the Avon Longitudinal Study of Parents and Children (ALSPAC). We are extremely grateful to all the families who took part in this study, the midwives for their help in recruiting them and the whole ALSPAC team, which includes interviewers, computer and laboratory technicians, clerical workers, research scientists, volunteers, managers, receptionists and nurses.

**Collaborators** Dr Penny Dobson provided advice on the strategies used by parents to overcome bedwetting and on the current NICE guidelines.

**Contributors** CJ, JH, MTG, KT and AW conceptualised and designed the study. MTG carried out the statistical analysis and drafted the Results section. JH, CJ and KT supervised the statistical analysis and interpreted the results. All authors were involved in drafting the manuscript and approved the final manuscript as submitted.

**Funding** This work was supported by the Medical Research Council grant number MR/L007231/1. The UK Medical Research Council, the Wellcome Trust (grant reference: 102215/2/13/2) and the University of Bristol provide core support for ALSPAC.

**Competing interests** None declared.

**Patient consent** Detail has been removed from this case description/these case descriptions to ensure anonymity. The editors and reviewers have seen the detailed

information available and are satisfied that the information backs up the case the authors are making.

**Ethics approval** ALSPAC Law and Ethics committee and local research ethics committees.

**Provenance and peer review** Not commissioned; externally peer reviewed.

**Data sharing statement** This is a secondary data analysis based on data from the ALSPAC cohort. The ALSPAC Executive encourages and facilitates data sharing with all 'bona fide' researchers. The access policy for the ALSPAC data can be found at http://www.bristol.ac.uk/media-library/sites/alspac/documents/ALSPAC_access_policy.pdf.

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
