## [Reviewer comments · BMJ Open]

ARTICLE DETAILS

TITLE (PROVISIONAL)	Examining the effectiveness of parental strategies to overcome bedwetting: an observational cohort study
AUTHORS	Grzeda, Mariusz; Heron, Jon; Tilling, Kate; Wright, Anne; Joinson, Carol

VERSION 1 - REVIEW

REVIEWER	Israel Franco, MD Director of the Yale/New Haven Children's Bladder and Continence Program Department of Urology Yale University New Haven CT, USA
REVIEW RETURNED	25-Mar-2017

GENERAL COMMENTS	The authors state that nocturnal enuresis is the new ICCS terminology for bedwetting but in actuality the new document uses the term enuresis only please correct this The article reads more like a statistical article to prove the value of PSM. since this is not a common technique used it would be useful if examples were used to outline the meaning of positive and negative results in the notes under the tables. There is no indication in the tables which is a positive result so we have we have to rely completely on the discussion to sort this out. I would be useful to mark positives in bold or add an * . It is unclear how the authors can say that lifting and restricting drinks is associated with increased risk of bedwetting if they could not control for the possibility that these children were the more severe wetters. if this is really the case then the authors need to state this immediately after this comment on page 19 para 1. the authors comments on lifting on page 20 are based off large cohort studies such as this but there is not scientific proof that these comments are accurate that is "it prevents the acquisition of dryness and the children are not learning bladder fullness" this again is based on conjecture from population studies that are incapable of separating out the more severe wetters from less severe wetters. repeating such statements just continues to propagate someones personal biases that are not founded in fact. The statement that there is evidence that restricting fluids can lead to reducing the ratio of bladder volume to overnight urine production is based on an article from 1984 that quotes articles from 1963-1983 which is clearly not what one would call solid modern proof especially what we understand today about the bladder and afferent
---

	signaling and central processing. this statement should be removed and the reference deleted, Our present knowledge does not support this statement and to reiterate it is again leading to a propagation of inaccurate data (or as our president and his cronies would call alternative facts). i just wish that i had a better description of the true positive results in the tables that would help the interpretation of the article.
--	--

REVIEWER	Sukanya De Consultant Paediatrician Sydney Australia
REVIEW RETURNED	09-Apr-2017

GENERAL COMMENTS	This study has a robust dataset and sample size with a strong potential for findings that could have clinical relevance. However some clarification is needed and the authors could present additional information (see below) that would be of use to clinicians: 1. Is the research question or study objective clearly defined?  • Since these are children who have enuresis, suggest rewording- the title from “to prevent bed wetting” to “to overcome bedwetting”; similarly the aim of the study would be to see if a range of common strategies “help overcome” rather than “reduce risk” of bedwetting. 3. Is the study design appropriate to answer the research question? 7. If statistics are used are they appropriate and described fully  • Given that this reviewer is not familiar with propensity scores, they are not in a position to comment on the choice and method of statistical analysis. 10. Are they (results) presented clearly?  • Table one should be referred to in results and not in methods. • Table 2: please explain tetrachoric correlation coefficients. • Propensity scores: The authors have stated that using propensity scores two of the parental strategies were associated with increased risk if bed wetting. A brief explanation of propensity scores/risk difference and the clinical relevance of a score/risk difference of 0.11 or 0.12 in the methods will be needed for clarity since most readers (including this reviewer) would not be familiar with this statistical tool. While the authors have explained propensity scoring in the supplementary material it would be more reader friendly to have a brief explanation of the relevance of a difference of 0.11 /0.12 in the methods as they have interpreted these scores as increased risk where as a score of <0.1 is described by them as negligible risk in the supplementary sections. Are there defined cut off scores for negligible versus substantial risk? 11. Are the discussion and conclusions justified by the results?  • See point 10 above. Not sure of the clinical significance of the propensity scores to state that there is an increased risk associated with two of the strategies. Additional comment:  • While confounding of effects of treatment due to other variables is probable, the treated and non treated groups are likely to be representative of the general population and hence it would still be of interest to know what percentage of children who had exposure to one or more parental strategies stopped bed wetting by age nine and half versus those who did not have exposure to any of the strategies. Then perhaps also looking at whether the “treated” and “not treated” groups differed significantly in terms of the potential confounders would be useful clinical information....it may be
--

	hypothesized that children who were treated with one or more strategies were different to those not treated in terms of presence/absence of behavioural challenges, soiling, day time urinary symptoms, socioeconomic characteristics. This has clinical relevance in terms of how parents approach the problem of bed wetting for example whether parents of children with behavioural/socioeconomic/day time bowel-bladder symptoms are less or more likely to attempt any strategies to overcome enuresis.
--	---

REVIEWER	Richard McNally Newcastle University, United Kingdom
REVIEW RETURNED	28-Apr-2017

GENERAL COMMENTS	GENERAL COMMENTS This is an interesting paper. However, there is a lack of detail in places and this requires attention. The cohort seems to have a high proportion of more affluent (higher socio-economic status) people. Specific comments are given below. SPECIFIC COMMENTS 1. Abstract. The interpretation of positive or negative differences is not clear. This needs careful and clear explanation. 2. The cohort seems to have a high proportion of more affluent (higher socio-economic status) people? Does this truly reflect the general population? Were there differences by social class? Do the results apply to the whole of the population? 3. Table 2. How can the correlation coefficients be interpreted? This requires clear explanation for a general readership. What is a high correlation? What is a low correlation? Give some numerical guidance. 4. Results. In Table 3 make clear exactly how positive and negative differences can be interpreted, i.e. clearly state that positive differences imply increased risk of bedwetting. 5. Discussion. Can the findings be generalised? Are there any caveats?
---

VERSION 1 – AUTHOR RESPONSE

Reviewer: 1

Reviewer Name: Israel Franco, MD

Institution and Country: Director of the Yale/New Haven Children's Bladder and Continence Program, Department of Urology, Yale University, New Haven CT, USA

Competing Interests: none declared

The authors state that nocturnal enuresis is the new ICCS terminology for bedwetting but in actuality the new document uses the term enuresis only please correct this

Response: Thank you. We have now changed 'nocturnal enuresis' to 'enuresis' throughout the manuscript.

The article reads more like a statistical article to prove the value of PSM. since this is not a common technique used it would be useful if examples were used to outline the meaning of positive and negative results in the notes under the tables. There is no indication in the tables which is a positive result so we have we have to rely completely on the discussion to sort this out. I would be useful to

mark positives in bold or add an * .

Response: As requested, we now provide a detailed description of how to interpret the meaning of the positive and negative results from table 3 in the footnote (please see page 18 of the revised manuscript). We have also marked the positive results (i.e. where there is evidence for an increase in bedwetting) in table 3 with a * .

It is unclear how the authors can say that lifting and restricting drinks is associated with increased risk of bedwetting if they could not control for the possibility that these children were the more severe wetters. If this is really the case then the authors need to state this immediately after this comment on page 19 para 1.

Response: At the top of page 22 we state: "We did, however, include a range of covariates in the analyses including frequency of bedwetting (high frequency= twice or more/week), daytime wetting, urgency and voiding postponement. All of these variables were important confounders in most of the models." Children who are bedwetting twice or more per week would be considered as having more severe bedwetting. Additionally, children with accompanying daytime symptoms might be considered as being more severe wetters.

The authors' comments on lifting on page 20 are based off large cohort studies such as this but there is not scientific proof that these comments are accurate that is "it prevents the acquisition of dryness and the children are not learning bladder fullness" this again is based on conjecture from population studies that are incapable of separating out the more severe wetters from less severe wetters. Repeating such statements just continues to propagate someone's personal biases that are not founded in fact.

Response: We agree that lifting requires further investigation in studies that are able to distinguish between more severe and less severe bedwetting and we have revised this paragraph (please see page 22 of the revised article).

The statement that there is evidence that restricting fluids can lead to reducing the ratio of bladder volume to overnight urine production is based on an article from 1984 that quotes articles from 1963-1983 which is clearly not what one would call solid modern proof especially what we understand today about the bladder and afferent signaling and central processing. This statement should be removed and the reference deleted. Our present knowledge does not support this statement and to reiterate it is again leading to a propagation of inaccurate data (or as our president and his cronies would call alternative facts).

Response: We have removed this statement and deleted the reference.

I just wish that I had a better description of the true positive results in the tables that would help the interpretation of the article.

Response: We thank the reviewer for drawing this to our attention and we feel that the description we now provide below table 3 should help the reader to interpret the meaning of the risk differences.

Reviewer: 2

Reviewer Name: Sukanya De

Institution and Country: Consultant Paediatrician, Sydney, Australia

Competing Interests: None Declared

This study has a robust dataset and sample size with a strong potential for findings that could have clinical relevance. However some clarification is needed and the authors could present additional information (see below) that would be of use to clinicians:

1. Is the research question or study objective clearly defined?

- Since these are children who have enuresis, suggest rewording- the title from “to prevent bed wetting” to “to overcome bedwetting”; similarly the aim of the study would be to see if a range of common strategies “help overcome” rather than “reduce risk” of bedwetting.

Response: We have revised the manuscript throughout using the term “overcome” rather than “prevent”.

3. Is the study design appropriate to answer the research question? 7. If statistics are used are they appropriate and described fully

- Given that this reviewer is not familiar with propensity scores, they are not in a position to comment on the choice and method of statistical analysis.

10. Are they (results) presented clearly?

- Table one should be referred to in results and not in methods.

Response: We have moved table 1 to the results section.

- Table 2: please explain tetrachoric correlation coefficients.

Response: We have added a sentence to the footnote of table 2 to explain that tetrachoric correlation coefficients are correlation coefficients of binary variables.

A detailed explanation is provided in: <http://www.stata.com/manuals13/rtetrachoric.pdf>

- Propensity scores: The authors have stated that using propensity scores two of the parental strategies were associated with increased risk if bed wetting. A brief explanation of propensity scores/risk difference and the clinical relevance of a score/risk difference of 0.11 or 0.12 in the methods will be needed for clarity since most readers (including this reviewer) would not be familiar with this statistical tool. While the authors have explained propensity scoring in the supplementary material it would be more reader friendly to have a brief explanation of the relevance of a difference of 0.11 /0.12 in the methods as they have interpreted these scores as increased risk where as a score of <0.1 is described by them as negligible risk in the supplementary sections. Are there defined cut off scores for negligible versus substantial risk?

Response: This comment relates to the request from reviewer 1 to provide an explanation of how to interpret the meaning of the results in table 3. We have now provided a detailed footnote below table 3 explaining how to interpret the risk differences (please see page 18 of the revised manuscript).

There are no defined cut offs for negligible versus substantial risk except the standard procedure of testing for statistical significance. The results provide evidence for an increased risk of bedwetting when the entire 95% confidence interval for the risk difference is above zero. When the lower bound of the 95% confidence interval lies below zero, the results cannot rule out the possibility that the parental strategy could also be associated with a reduced risk of bedwetting i.e. the results are consistent with either an increase or a decreased risk of bedwetting.

The risk difference can be thought of as a measure of the excess risk of disease in those who have the risk factor (i.e. parents who used the particular strategy to overcome bedwetting) compared with those who don't. For instance, the excess risk of bedwetting in children whose parents restricted drinks is 12.3% (between 2.1% and 22.6%).

11. Are the discussion and conclusions justified by the results?

- See point 10 above. Not sure of the clinical significance of the propensity scores to state that there is an increased risk associated with two of the strategies.

Response: We hope that our response to the comment above has clarified the interpretation of the results, particularly where there is evidence of an increased risk of bedwetting associated with the strategy.

Additional comment:

- While confounding of effects of treatment due to other variables is probable, the treated and non treated groups are likely to be representative of the general population and hence it would still be of interest to know what percentage of children who had exposure to one or more parental strategies stopped bed wetting by age nine and half versus those who did not have exposure to any of the strategies. Then perhaps also looking at whether the “treated” and “not treated” groups differed significantly in terms of the potential confounders would be useful clinical information....it may be hypothesized that children who were treated with one or more strategies were different to those not treated in terms of presence/absence of behavioural challenges, soiling, day time urinary symptoms, socioeconomic characteristics. This has clinical relevance in terms of how parents approach the problem of bed wetting for example whether parents of children with behavioural/socioeconomic/day time bowel-bladder symptoms are less or more likely to attempt any strategies to overcome enuresis.

Response:

We identified three main questions within the additional comment above and we have addressed each one in turn:

(1) While confounding of effects of treatment due to other variables is probable, the treated and non treated groups are likely to be representative of the general population

Response: The propensity score methods we used are aimed at making the treated and non-treated groups as similar as possible, therefore, minimising confounding. We discuss this in detail in the supplementary materials submitted with our manuscript. Briefly, propensity score-based methods can be used to minimise effects of confounders when estimating treatment effects using observational data. These methods adjust the crude results to take into account the non-random mechanism of selection of cases to the ‘treated’ versus ‘non-treated’ groups by controlling for differences between both groups. Propensity score-based methods are based on the assumption that differences between ‘treated’ and ‘non-treated’ groups can be explained in terms of observable variables and bias should, therefore, be minimised by ensuring similarity of both groups on these variables. This is achieved using propensity score-based methods either by matching each case from the ‘treated’ group to at least one similar case from the ‘non-treated’ group or by re-weighting both groups in order to construct a synthetic sample in which both groups are similar with respect to the observed confounders. Similarity of the groups is assessed in terms of the probability of receiving the ‘treatment’ as estimated from the observed variables of the cases. This probability, called the ‘propensity score’, is further used to create/or reshape the sample in which distributions of variables of study participants are similar in both groups. If this condition is fulfilled then treated and untreated groups are balanced on measured confounders.

(2) and hence it would still be of interest to know what percentage of children who had exposure to one or more parental strategies stopped bed wetting by age nine and half versus those who did not have exposure to any of the strategies.

We think the reviewer is referring to the crude; unadjusted estimates of the percentage of children

who had exposure to one or more parental strategies and stopped bedwetting by age 9½ versus those who did not have exposure to any of the strategies. These unadjusted results are likely to be biased because they are not adjusted for potential confounders (see above). The results we have provided in the manuscript, based on the propensity score methods, compare the risk of bedwetting in the treated and non-treated groups balanced for a wide range of confounders.

(3) Then perhaps also looking at whether the “treated” and “not treated” groups differed significantly in terms of the potential confounders would be useful clinical information...it may be hypothesized that children who were treated with one or more strategies were different to those not treated in terms of presence/absence of behavioural challenges, soiling, day time urinary symptoms, socioeconomic characteristics. This has clinical relevance in terms of how parents approach the problem of bed wetting for example whether parents of children with behavioural/socioeconomic/day time bowel-bladder symptoms are less or more likely to attempt any strategies to overcome enuresis.

In the supplementary material (part (d) Diagnostics), we explain how we carried out checks to examine whether treated and non-treated groups are similar in terms of the confounders. The confounders (listed in table S1) include the areas mentioned by the reviewer (behavioural difficulties, soiling, daytime urinary symptoms, and a range of socioeconomic characteristics). Briefly, these diagnostic methods are designed to examine whether the distributions of the confounding variables are the same/or similar enough in treated and non-treated groups. Before analysing the treatment effects, we assessed the adequacy of the propensity score model by checking the overlap in propensity scores. We based our diagnostic checks on the examination of differences between means of baseline variables in the weighted sample and we found our models to be adequate.

Reviewer: 3

Reviewer Name: Richard McNally

Institution and Country: Newcastle University, United Kingdom

Competing Interests: None declared

GENERAL COMMENTS

This is an interesting paper. However, there is a lack of detail in places and this requires attention. The cohort seems to have a high proportion of more affluent (higher socio-economic status) people. Specific comments are given below.

SPECIFIC COMMENTS

1. Abstract. The interpretation of positive or negative differences is not clear. This needs careful and clear explanation.

Response: We have added an explanation of how to interpret the risk difference in the abstract and in the footnote for table 3 (please see page 18 of the revised manuscript).

2. The cohort seems to have a high proportion of more affluent (higher socio-economic status) people? Does this truly reflect the general population? Were there differences by social class? Do the results apply to the whole of the population?

Response: Details about the representativeness of the ALSPAC cohort are provided at <http://www.bristol.ac.uk/alspac/researchers/cohort-profile/representativeness/>. Briefly, mothers of infants in Avon were slightly more likely than those in the rest of Britain to live in owner occupied accommodation, to have a car available to the household and to be less likely to have one or more persons per room and be non-white. Thus, similar to all studies where a representative sample has been attempted, this study had a slight shortfall in the less affluent families (those living in rented

accommodation, not having a car or being single or unmarried cohabiting). The study also had a shortfall in ethnic minority mothers. These slight differences between the cohort and the whole of Great Britain are unlikely to explain the results we have found. Additionally, we have previously found no evidence for an association between bedwetting and social class in the cohort.

The propensity score model included a range of socioeconomic indicators (please see table S1), therefore, the results are unlikely to differ by social class.

3. Table 2. How can the correlation coefficients be interpreted? This requires clear explanation for a general readership. What is a high correlation? What is a low correlation? Give some numerical guidance.

Response: In response to the comment from reviewer 2 about tetrachoric correlation coefficients, we have added a sentence to the footnote of table 2 to explain that tetrachoric correlation coefficients are correlation coefficients of binary variables. In response to the current reviewer's comment we have added a further explanation to the footnote explaining how to interpret these correlation coefficients:

"The tetrachoric correlation coefficient provides an estimate of what the correlation would be if the variables were measured on a continuous scale. The size of the tetrachoric correlation coefficient can be interpreted in the same way as a Pearson's coefficient of linear correlation between 2 continuous variables i.e. '0' indicates no correlation and '1' indicates perfect correlation."

A detailed explanation is provided in: <http://www.stata.com/manuals13/rtetrachoric.pdf>

4. Results. In Table 3 make clear exactly how positive and negative differences can be interpreted, i.e. clearly state that positive differences imply increased risk of bedwetting.

Response: This comment relates to comments from the first two reviewers and we have now added a detailed explanation to the footnote for table 3 to explain how to interpret the positive and negative risk differences (please see page 18 of the revised manuscript).

5. Discussion. Can the findings be generalised? Are there any caveats?

Response: In terms of external validity, we believe that the results could potentially be generalised to the rest of the UK (please see our response to comment 2 above) and probably generalised to other countries where parents use similar strategies to overcome bedwetting and where there are similar confounding factors. A potential caveat affecting internal validity is selective attrition. In ALSPAC drop out is related to socioeconomic disadvantage, with more disadvantaged families dropping out of the study. Selection bias is an unlikely explanation for our findings because bedwetting is not related to socioeconomic disadvantage in the ALSPAC cohort.

VERSION 2 – REVIEW

REVIEWER	Richard McNally Institute of Health & Society, Newcastle University, United Kingdom
REVIEW RETURNED	15-May-2017

GENERAL COMMENTS	The authors have addressed all of my concerns.
--